# Factors influencing exclusive breast feeding among children born to HIV positive mothers attending public health facilities in western Ethiopia: Cross-sectional study

Ejigayehu Tolessa Bultum[1,2☯], Elias Merdassa Roro[1☯]*, Tsedeke Wolde[1,3†‡], Ilili Feyessa Regasa[1‡]

1 Department of Public Health, Institute of Health Sciences, Wollega University, Nekemte, Oromia, Ethiopia, 2 Federal Democratic Republic of Ethiopia, Ministry of Health, Addis Ababa, Ethiopia, 3 Federal Democratic Republic of Ethiopia, Ministry of Health, St Paul Millennium Medical college, Addis Ababa, Ethiopia

☯ These authors contributed equally to this work.
† Deceased.
‡ TW and IFR also contributed equally to this work.
* emerdassa@gmail.com, eliasm@wollegauniversity.edu.et

**Data Availability Statement:** All relevant data are within the manuscript and its Supporting Information files.

## Abstract

Only about 39% of infants in low- and middle-income countries are exclusively breast-fed for the first six months. In particular, human immunodeficiency virus (HIV) positive women report confusion about the best feeding methods. Exclusive Breastfeeding (EBF) practices in HIV positive mothers are sub-optimal in Ethiopia. This study aimed to identify the main factors influencing EBF among HIV positive breast-feeding mothers. A facility based cross-sectional study was carried out from September 2017 to June 30 2018 among HIV positive mothers with infants aged 6–23 months. Thirteen public health facilities (7 health centers and 6 hospitals) that provided anti-retroviral treatment (ART) and Prevention of mother-to-child transmission (PMTCT) services, found in three districts of West, East and Kellem Wollega Zones, were randomly selected. Respondents were recruited by systematic random sampling techniques from these facilities using client registers as a sampling frame. Data were collected using face to face interviews with a pre-tested questionnaire. Data were entered into EPI info Version 3.5.1 and analyzed using SPSS Version 20 for windows. Candidate variables for the final multi-variable model were selected considering $P \leq 0.05$ at bivariable analysis. Associations were declared at $P \leq 0.05$ by assuming Confidence Intervals did not cross '1'with corresponding 95%. A total of 218 HIV positive mothers were included in this study. Of these, only 122 (56.0%) practiced EBF in the first six months. The proportion of study participants who initiated EBF within the first hour of delivery was 134 (61.8%). Mean age of study participants was 28.6years with SD ± 4. Mothers' having received advice on EBF [AOR 3, 95% CI (1.2–6.7)], disclosure of HIV status to someone close to them including their husband [AOR 6, 95% CI (1.2–29.6)] and believing HIV can be transmitted during delivery [AOR 5.2, 95% CI (1.1–24.0)] were found to increase the likelihood of EBF practices among the study participants (P-value ≤ 0.05). In this study, only just over half of the mothers practiced EBF for the first six months. Care providers should

**Funding:** The authors received no specific funding for this work.

**Competing interests:** The authors have declared that no competing interests exist.

continue to encourage mothers to practice EBF in the first six months and to disclosure of HIV status to someone close to them including their partner. Efforts should be in place to curb the risk of HIV/AIDS transmission during delivery. Continues advise for mothers to practice EBF in the first 6 months is still needed.

## Background

The risk of mother-to-child transmission (MTCT) of Human Immunodeficiency virus (HIV) through breastfeeding is challenging though it is an accepted traditional way of promoting health for infants and children in developing countries [1]. MTCT of HIV remains the most significant route of HIV infection among children [2]. Annually, 700,000 infants acquire HIV infection from their mothers where, 280,000 of which are infected through breastfeeding [3]. In the absence of interventions during pregnancy and delivery, HIV transmission through breastfeeding could be responsible for as high as a third of all childhood HIV infections [4].

Despite this, the World Health Organization (WHO) recommends Exclusive Breast Feeding (EBF) by HIV positive mothers during the postnatal period in low- and middle-income countries. The high cost of Exclusive Replacement Feeding (ERF) and lack of adequate clean water and poor sanitation [5, 6] means ERF is not feasible or safe in low- and middle-income countries. While breast milk can be a source of HIV, the risk to infants, particularly those in low-income settings, of malnutrition, diarrhea, acute respiratory tract infection, and death if they are not breastfed remains higher [7]. Furthermore, EBF for up to six months is a more effective strategy for reducing the risk of HIV transmission 3-4-fold compared to mixed feeding [6]. The new recommendation from the WHO regarding commentary feeding for the first 12 months found to be a better strategy [8] than the earlier versions of the strategy.

In higher income contexts, the avoidance of breast feeding by HIV positive mothers if replacement is available is acceptable, feasible, affordable, sustainable and safe (AFASS) [6].

In line with the WHO recommendations, the Federal Ministry of Health of Ethiopia has adopted and developed a strategy of infant and young child feeding. According to the national strategy, informed choices that suit the circumstances of the mother are emphasized and the advice for mothers should be tailored to their individual needs to balance the risks associated with replacement feeding with the risks of contracting HIV via breast feeding [4].

However, it should be noted that controversy remains over the optimum time to introduce commentary feeding [9], and women living with HIV/AIDS commonly report feeling confused over breast-feeding methods [5]. For example, studies have revealed a high rate of early cessation of breastfeeding because of the fear of transmitting HIV/AIDS to their babies [10–12]. Such practices have been associated with malnutrition, sudden infant death syndrome, other neonatal morbidity and mortality [13–16]. Such early cessation is common in the critical period for infant growth (4–6 months) among HIV/AIDS exposed children [14].

Though breastfeeding of infants born to HIV positive mothers remains a risk of acquiring HIV infection [17], there is evidence that EBF decreases the chances of HIV infection in exposed infants compared to mixed feeding. This is because Exclusive Replacement Feeding (ERF) is not feasible due to the high cost of supplementary feeds and lack of adequate clean water and poor sanitation in developing countries [5].

In Sub-Saharan Africa, the roll-out of both highly active anti-retroviral (ART) and more limited pre-natal ART, as well as improved understanding of risk factors for HIV transmission through breastfeeding have dramatically reduced the number of infants becoming HIV-infected

[18]. However, there is a paucity of studies, which show predictors of breastfeeding cessation among HIV positive mothers in Ethiopia. Identifying factors related to exclusive breastfeeding practices among HIV infected women is important for targeting evidence-based intervention, which helps to increased HIV-free survival.

This study focused on EBF practices among HIV positive mothers in a region of Ethiopia where there have been no previous well-established facility-based studies. The study outcomes can be used to help improve services for these women by identifying the main factors which enhance EBF practices among HIV positive mothers.

## Methods and materials

### The setting, study design, and populations

The study was conducted from September 2017 to June 2018 in selected health facilities found in East, West and Kellem Wollega districts. The three districts have a population of 4,322,357 with a mean of 1,440,786 in each district and a 1:1 gender ratio. In total there were 893,536 women of reproductive age group (15–45 years old) and about 710,163 under five children at the time of the study [19].

Seven health centers and six hospitals providing ART and PMTCT services were randomly selected for the study. Data were collected by 13 ART nurses from the respective health institutions. Three supervisors were recruited to oversee data collection. Across the 3 districts, an estimated 2,365 HIV positive women were on antiretroviral therapy (ART) with a total of 415 mothers with HIV/ exposed infants (HEI).

A cross-sectional study was conducted among 219 HIV positive mothers with children under six months and attending PMTCT and ART clinics in the selected public hospitals and health centers were randomly selected.

**Inclusion criteria.** Mothers living with HIV/AIDS with a child aged 6 months or less who voluntarily consented to participate in the study were included.

**Exclusion criteria.** Mothers living with HIV/AIDS with a child aged 6 months or less who did not consent to participate and/or who were seriously ill and unable to provide informed consent were excluded from the study.

### Sample size determination

The study was conducted in thirteen randomly selected government owned health institutions (six hospitals and seven health centers) proving ART and PMTCT services for mothers living with HIV/AIDS. These institutions were selected based on the availability of the aforementioned services and adequacy of client flow.

The sample size (n) required for the study was calculated using a single population proportion with assumptions of 95% confidence interval, 5% desired precision, proportion of HIV positive mothers exclusively breast feed their infants 48.2% [15] and considering 10% for compensation for non-response. The total samples calculated for the study was 219. The total sample size was across each of the 13 selected health care institutions using probability proportional to size (PPS) and the numbers of HIV positive mothers required for the study in each PMTCT/ART clinic was determined. For each selected facility, a sampling frame was prepared. Participants were then selected from their respective health facility using a systematic random sampling technique from the sampling frame. Sampling intervals were determined by dividing N/n (415/219), i.e., every second intervals. The first respondent was selected by blindly choosing one out of two pieces of paper numbered 1 & 2. Every second mothers visiting the health facility to collect ARV medication or for other purposes were recruited and interviewed.

## Data collection procedures

The interview questionnaire was prepared by reviewing relevant literatures as part of this study. The interview questionnaire contained eight (8) sections, each addressing different issues. The questionnaire was first developed in English and, translated into the local language (Afaan Oromo) then, back translated to English to check for its consistency by language experts who can understand both languages. The tool was pre-tested among breast feeding mothers to check language clarity, administration procedures and consistency in a similar setting to the study site, but outside of the study catchment area. After pre-testing, data were analyzed and carefully examined for any discrepancy between actual meaning and respondents' answers. After thorough discussion between study investigators, data collectors, and supervisors, some ambiguous words were modified, and misunderstandings were clarified. Data from the pre-test are not included in this manuscript.

The data in this study were collected by 13 ART nurses from the respective health institutions who were working at ART and PMTCT departments. Three supervisors and the principal investigator performed overall controlling activities of the data collections process. Two days intensive training outlining data collection process and tools was provided for data collectors and supervisors.

## Defining study variables

The dependent Variable was EBF. EBF is defined as the consumption of only breast milk with no supplementation of any type of food from birth except drops and syrups such as; vitamins, minerals and medicines [20]. For this study we dichotomized EBF into ('Yes' and 'No'). Those mothers who exclusively breast fed their children for the first six months were labeled as 'Yes.' Those mothers who started complementary feeding within the first six months were labeled as 'No.'

Independent Variables assessed were: Socio economic status of mothers and households, educational status of mothers, income of mothers, antenatal care attendance, occupation of mothers, disclosure of HIV status to spouse, age of mothers, parity and mode of delivery, mother's decision on the choice of infant feeding, infant illnesses, baby hospitalization, birth weight, hospital and health service, nutritional education of the mother, antenatal care, post-natal care, cultural norms on breastfeeding, child feeding practice and feeding system, mixed feeding (defined as breast -feeding with the addition of fluids, solid feeds and non-human milks in the first 6 months of age) [20].

HIV Exposed Infants were defined as an infant or child born to a mother living with HIV until the infant or child is reliably excluded from being HIV infected [21]. HIV positive mothers refers to women belonging to the age group of 15 to 49 years who are on ART, attending the antenatal and post-natal clinics in the selected hospitals and health centers.

Early termination of breast-feeding was defined as the act of interrupting giving breast milk and making the child accustomed to other food before two years of age [21]. Seriously ill was defined as women who are sick, in bed and unable to provide information.

## Data processing and management

Data were entered into EPI info Version 3.5.1 and transferred to SPSS Version 20 for windows statistical package for analysis. Results were reported using standard data presentation techniques including frequency tables, measures of central tendencies and variations. Both bi-variable and multivariable logistic regression analysis were used to determine the association of each independent variable with the dependent variable. Candidate variables for the multi-variable model were identified by considering $P \leq 0.05$ at bi-variable analysis. A multi-variable

logistic regression model was used to control for the effects of confounders on the outcome variable. Associations were declared if P≤0.05 with 95% confidence intervals and Adjusted Odds Ratio (AOR) reported to show the strength of associations.

### Ethics approval and consent to participate

This study was approved by the Wollega University, Institute of Health Sciences, Research Ethics Committee. Letters of permission were also obtained from East, West and Kellem wollega zones health offices and the respective health offices and facilities of the three selected districts and kebeles (the smallest administrative division of the government). A consent form was attached to every questionnaire with a brief information sheet explaining the aims of the study, benefits, and risks of the study to the participants, explaining their full right to withdraw at any time and their right to skip any question that they did not want to answer. Written consent was obtained from each participant. Data were collected after each participant signed a consent form. In cases where data were requested from minors, consent was sought from their parents or guardians. Only those who consented participated in the study. Data were collected at each selected health facility (i.e., hospital or health centers) in an area providing maximum privacy. All individual identifiers were removed to maintain anonymity of participants and each questionnaire was assigned a unique study number.

## Results and discussions

### Results

**Socio-demographic characteristics.** Of 219 randomly selected mothers with children aged ≤6 months, 218 participated in this study, a response rate of 99.5%. Of these, the majority of participants, 184 (84.4%) were married. The average age was 28.6 years with standard deviation (SD) of 4. A higher proportion, 139 (63.8%), of participants were protestant. Around 99 (44.0%) had completed grade 1–8. The majority, 194 (89.0%) were Oromo. More than half, 129 (59.2%) were housewives. The mean age of the infants was 6 months (SD ±2.8) and 113 (51.8%) were males (Table 1).

**Knowledge regarding the benefits of exclusive breast feeding.** Questions exploring participants' knowledge of MTCT, PMTCT and benefits of EBF found 179 (82.0%) of mothers were aware that HIV could be transmitted through breast milk. 62 (28.4%) mothers did not know that breast- feeding protected the infant from diarrhea. 50 (22.9%) mothers were aware that HIV could be transmitted from mother to child during pregnancy, delivery and breast-feeding.

**Obstetric history of participants.** Table 2 outline the obstetric history of respondents and the type of feeding advice obtained from health workers. More than half, 119 (54.6%), of mothers had attended more than four antenatal visits. The majority, 214 (98.2%), of the mothers delivered at health institutions and the mode of delivery was spontaneous vaginal delivery, 182 (83.5%). The majority 209 (95.9%) had attended postnatal care (PNC) follow up and 169 (77.5%) had received advice on infant feeding (Table 2).

**Proportions of HIV positive mothers exclusively breast feeding.** Of all respondents, only 122 (56.0%) practiced EBF. The proportion of participants who initiated EBF within the first hours of delivery was 134 (61.5%) while a few, 27(12.4%), initialed EBF twenty-four hours after birth (Table 3).

**Determinants of exclusive breast feeding.** After controlling for the effect of confounders, variables found to be statistically associated with EBF practices among HIV positive mothers were; exposure to feeding advice, disclosure of HIV status to close friends including husbands and mothers who were aware that HIV could be transmitted through delivery (p-value<0.05) (Table 4).

**Table 1. Socio-demographic characteristics of study participants (n = 218), West, East and Kellem Wollega Districts health institution west Ethiopia, September 2017 to June 30, 2018.**

| Variables | Frequency | Percent | P-values |
|---|---|---|---|
| Age of mothers (years) | | | 0.976 |
| 15–19 | 1 | 0.5 | |
| 20–24 | 32 | 14.7 | |
| 25–29 | 96 | 44.0 | |
| 30–34 | 66 | 30.3 | |
| >35 | 23 | 10.6 | |
| Age of child in months | | | 0.552 |
| 6–11 | 185 | 84.0 | |
| 12–18 | 33 | 16.0 | |
| Sex of child | | | 0.632 |
| Male | 113 | 51.8 | |
| Female | 105 | 48.2 | |
| Marital Status | | | 0.218 |
| Single | 11 | 5.0 | |
| Married (union) | 184 | 84.4 | |
| Divorced | 16 | 7.3 | |
| Widowed | 7 | 3.2 | |
| Educational status | | | 0.979 |
| Unable to read & write | 26 | 11.9 | |
| Able to read & write | 48 | 22.0 | |
| Grade 1–8 | 96 | 44.0 | |
| Grade 9–12 | 40 | 18.3 | |
| College & above | 8 | 3.7 | |
| Religion | | | 0.429 |
| Protestant Christian | 139 | 63.8 | |
| Others | 79 | 36.3 | |
| Ethnicity | | | 0.045 |
| Amara | 16 | 7.3 | |
| Oromo | 194 | 89.0 | |
| Others | 8 | 3.6 | |
| Maternal occupation | | | 0.492 |
| Self Employed | 44 | 20.2 | |
| House wife | 129 | 59.2 | |
| Merchant | 26 | 11.9 | |
| Farmer | 11 | 5.0 | |
| Government employee | 8 | 3.7 | |

Mothers' who received advice on EBF were three times more likely to practice EBF than those who were not given advice [AOR 3 (95% CI (1.2–6.7)]. Those who disclose their HIV status to their spouse were six times more likely to adhere to EBF [AOR 6, 95% CI (1.2–29.6)]. Those who were aware that HIV could be transmitted during delivery were more than five times more likely to adhere to EBF than who were unaware [AOR 5.2, 95% CI (1.1–24)] (Table 4).

## Discussion

In this study, we assessed factors influencing EBF among HIV positive mothers.

**Table 2. Obstetric history of study participants (n = 218) West, East and Kellem Wollega Districts health institutions western Ethiopia, September 2017 to June 30, 2018.**

| Variables | Frequency | Percent | P-values |
|---|---|---|---|
| No of ANC follow up | | | 0.091 |
| Two | 17 | 7.8 | |
| Three | 71 | 33.6 | |
| Four and above | 119 | 54.6 | |
| Don't remember | 11 | 5.0 | |
| Place of delivery | | | 0.337 |
| Home | 4 | 1.8 | |
| Health institution | 214 | 98.2 | |
| Type of Delivery | | | 0.023 |
| Normal | 182 | 83.5 | |
| Caesarian Section | 36 | 16.5 | |
| Post-natal follow up | | | 0.262 |
| Yes | 209 | 95.9 | |
| No | 9 | 4.1 | |
| Feeding advices obtained from HW | | | 0.024 |
| No | 49 | 22.5 | |
| Yes | 169 | 77.5 | |

The proportion of mothers with children less than six months who exclusively breast fed their children in this study (56%) was lower than a similar study conducted in Addis Ababa [22]. This may be explained by the different geographical contexts. In Addis Ababa, mothers may be exposed to better media coverage promoting EBF and may have better access to counseling, MTCT and PMTCT services than in our current study setting. Compared to our findings, another study conducted in Tigray region reported a substantially higher prevalence of EBF of 90% [23]. The difference may be explained by methodological variations between studies, socio-cultural factors and health service utilization characteristics among participants in the different study areas.

**Table 3. Exclusively breast-feeding practice among HIV positive mothers in west, east and kellem Wollega Districts health institutions western Ethiopia, September 2017 to June 30, 2018.**

| Variables | Frequency | Percent | P-values |
|---|---|---|---|
| **Exclusive Breast Feeding** | | | |
| Yes | 122 | 56.0 | |
| No | 96 | 44.0 | |
| **Do you give colostrum to new born infant** | | | 0.257 |
| Yes | 168 | 77.1 | |
| No | 50 | 22.9 | |
| **When have you started breast feeding your child?** | | | 0.128 |
| Within one hour after birth | 134 | 61.8 | |
| Within two hours after birth | 55 | 25.3 | |
| Within 24 hours after birth | 27 | 12.4 | |
| After one day | 2 | 0.9 | |
| **Reason for not giving first milk/ colostrum** | | | 0.584 |
| Her breast has no milk | 26 | 52.0 | |
| The mother is sick | 23 | 46.0 | |
| Health worker advised not to give | 1 | 0.5 | |

**Table 4. Determinants of Exclusive Breast-feeding Adherence among HIV Positive Mothers in West, East and Kellem Health institutions, western Ethiopia, September 2017 to June 30, 2018.**

| Variables | Exclusive breast-feeding status | | COR (95% CI) | AOR (95% CI) |
|---|---|---|---|---|
| **Age of mothers** | **No (n = 96)** | **Yes (n = 122)** | | |
| 15–19 | 1(0.8%) | 1(0.8%) | 0.9(0.7–1.6) | 0.4(0.2–9.5) |
| 20–24 | 21(21.9%) | 11(9.0%) | 0.3(0.1–0.9) | 0.3(0.1–9.9) |
| 25–29 | 39(40.6%) | 57(46.7%) | 0.8(0.3–2.0) | 0.5(0.1–17.9) |
| 30–34 | 28(29.2%) | 38(31.1%) | 0.7(0.3–1.9) | 0.3(0.1–3.5) |
| 35> | 8(8.3%) | 15(12.3%) | 1 | 1 |
| **Child feeding advice obtained from HWs** | | | | |
| No | 33(34.4%) | 15(12.3%) | 1 | 1 |
| Yes | 63(65.6%) | 107(87.7%) | **3.7(1.9–7.4)** | **3(1.20–6.7)** * |
| **Infant Birth Weight** | | | | |
| ≥2.5kg | 3(5.5%) | 4(4.4%) | 1 | 1 |
| <2.5kg | 52(94.5%) | 87(95.6%) | **1.9(1.1–3.3)** | 1.4(0.7–2.6) |
| **HIV disclosure status** | | | | |
| No | 5(5.2%) | 6(4.9%) | 1 | 1 |
| Yes | 91(94.8%) | 116(95.1) | **0.1(0.3–0.6)** | **6 (1.2–29.6)** * |
| **When HIV transmitted** | | | | |
| During delivery | 5(5.2%) | 17(13.9%) | **3.2(1.9–11.0)** | **5.2(1.1–24.4)** * |
| During pregnancy | 13(13.5%) | 14(11.5%) | 0.8(0.4–1.8) | 0.8(0.3–1.8) |
| During Breast feeding | 62(64.6%) | 50(41.0%) | 2.4(0.9–6.3) | 2.3(0.8–6.8) |
| I don't know | 2(2.1%) | 5(4.1%) | 1 | 1 |

*Associated at P≤ 0.05

This study identified important factors related to EBF. Mothers who received advice on child EBF were three times more likely to exclusively breast feed their children than those who were not. A systematic review from low- and middle-income countries concluded that improving the counselling skills of health workers to address breastfeeding problems is a critical component of infant and young child feeding programmes, and would aid in attaining the 2025 World Health Assembly EBF targets [24]. Another systematic review and meta-analysis from Ethiopia reported similar findings [25]. According to the Federal Ministry of Health in Ethiopia, all HIV-infected mothers should receive counseling which includes provision of general information about the risks and benefits of various infant feeding options [26].

Mothers who disclose their HIV status to someone close to them, including their husband, were 6 times more likely to adhere to EBF for the first six months than their counter parts. This finding is consistent with a systematic review and meta-analysis from Ethiopia [25], a study from south Ethiopia [27] and the findings of a similar study conducted in Gondar which showed disclosure of HIV status was independently associated with EBF [11]. The high disclosure rate among HIV positive mothers appears to be strongly associated with the practice of EBF in these mothers [28]. Those who were aware that HIV could be transmitted during delivery were 5.2 times more likely to adhere to EBF than those who did not know the mechanisms of HIV transmission from mother to child.

Our study had some limitations. Our findings need to interpreted with caution as other confounding factors may have impacted when establishing associations as we considered only those variables with P-values <0.05 at bi-variable analysis. We also had limited variables included in the final model.

## Conclusion

This study determined that only slightly more than half of the mothers living with HIV/AIDS who had children aged six months or less practiced EBF in the study area. The major factors identified favoring EBF were; being advised by a health extension worker about appropriate child feeding practices, disclosure of HIV status to someone close to them (including their husband) and knowing that HIV can be transmitted during delivery.

## Recommendations

Since EBF is a less costly and more practical option in the context of resource constrained countries; health workers should give advice on MTCT, PMTCT and infant feeding practice during PNC follow up to improve the knowledge of mothers in this area. Furthermore, supporting mothers to disclose HIV status to someone close to them (including their husband) by encouraging them to bring them to the health facility and be party to discussions with health care workers may be beneficial.

## Limitation of the study

Since the study design was cross sectional, it only explored the association of breast-feeding practices with other explanatory factors. There might be recall biases amongst participants who were reporting retrospectively.

## Supporting information

**S1 Data.**
(SAV)

## Acknowledgments

We thank all participants, data collectors, and field workers for their genuine co-operation. We are also delighted to acknowledge East, West, and Kellem wollega district health offices for their unreserved assistance from the start until the completion of this study. We appreciate Clare Phillips and Christine Andrews for their contributions in language and readability edition of the manuscript. "Tsedeke Wolde passed away before the submission of the final version of this manuscript. Elias Merdassa Roro [Corresponding author] accepts responsibility for the integrity and validity of the data collected and analyzed.

## Author Contributions

**Conceptualization:** Ejigayehu Tolessa Bultum, Elias Merdassa Roro, Tsedeke Wolde.

**Data curation:** Ejigayehu Tolessa Bultum, Elias Merdassa Roro.

**Formal analysis:** Ejigayehu Tolessa Bultum, Elias Merdassa Roro, Tsedeke Wolde.

**Funding acquisition:** Ejigayehu Tolessa Bultum.

**Investigation:** Ejigayehu Tolessa Bultum, Elias Merdassa Roro, Tsedeke Wolde.

**Methodology:** Ejigayehu Tolessa Bultum, Elias Merdassa Roro, Tsedeke Wolde, Ilili Feyessa Regasa.

**Project administration:** Ejigayehu Tolessa Bultum.

**Resources:** Ejigayehu Tolessa Bultum, Elias Merdassa Roro, Tsedeke Wolde, Ilili Feyessa Regasa.

**Software:** Ejigayehu Tolessa Bultum, Elias Merdassa Roro, Tsedeke Wolde, Ilili Feyessa Regasa.

**Supervision:** Ejigayehu Tolessa Bultum, Elias Merdassa Roro.

**Validation:** Ejigayehu Tolessa Bultum, Elias Merdassa Roro, Tsedeke Wolde, Ilili Feyessa Regasa.

**Visualization:** Ejigayehu Tolessa Bultum, Elias Merdassa Roro, Ilili Feyessa Regasa.

**Writing – original draft:** Ejigayehu Tolessa Bultum, Elias Merdassa Roro, Tsedeke Wolde, Ilili Feyessa Regasa.

**Writing – review & editing:** Ejigayehu Tolessa Bultum, Elias Merdassa Roro, Ilili Feyessa Regasa.

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
