## [Decision Letter · Decision Letter 0]

3 Sep 2021

PONE-D-21-08205

Status of Exclusive Breast Feeding among children born to Human immunodeficiency virus (HIV) positive mothers attending public health facilities in western Ethiopia. Cross-sectional study.

PLOS ONE

Dear Mr Elias Roro,

Thank you for submitting your manuscript to PLOS ONE. After careful consideration, we feel that it has merit but does not fully meet PLOS ONE’s publication criteria as it currently stands. Therefore, we invite you to submit a revised version of the manuscript that addresses the points raised during the review process.

We look forward to receiving your revised manuscript.

Kind regards,

Professor Kwasi Torpey, MD PhD MPH

Academic Editor

PLOS ONE

Journal Requirements:

3. You indicated that you had ethical approval for your study. In your Methods section, please ensure you have also stated whether you obtained consent from parents or guardians of the minors included in the study or whether the research ethics committee or IRB specifically waived the need for their consent.

4. Please ensure you have thoroughly discussed any additional potential limitations of this study within the Discussion section, including the potential impact of confounding factors.

7. We suggest you thoroughly copyedit your manuscript for language usage, spelling, and grammar. If you do not know anyone who can help you do this, you may wish to consider employing a professional scientific editing service.

Reviewers' comments:

Reviewer's Responses to Questions

**Comments to the Author**

1. Is the manuscript technically sound, and do the data support the conclusions?

Reviewer #1: Partly

Reviewer #2: Yes

2. Has the statistical analysis been performed appropriately and rigorously? 

Reviewer #1: No

Reviewer #2: Yes

3. Have the authors made all data underlying the findings in their manuscript fully available?

Reviewer #1: Yes

Reviewer #2: Yes

4. Is the manuscript presented in an intelligible fashion and written in standard English?

Reviewer #1: No

Reviewer #2: No

5. Review Comments to the Author

Reviewer #1: General comments

• This is a good topic and will be of great interest to readers

• The manuscript needs to be edited for grammar and syntax (by a native English speaker)

Major comments

Topic: Status of Exclusive Breast Feeding among children born to HIV positive mothers attending public health facilities in western Ethiopia. Cross-sectional study.

• This study is more of Exploring factors associated with or influencing EBF among children born to HIV positive mothers and not status of EBF as stated by the authors

Abstract

• Page 2: The abstract of this paper should contain a statement on the background of EBF among HEIs in Ethiopia. This is missing.

• The purpose the of the study is not consistently stated throughout the paper. The authors claim to be assessing the levels and factors influencing EBF among HEIs while the topic reads Status of EBF. The study should have one clear focus and purpose that’s carried out throughout.

• The proportion of respondents who initiated Exclusive Breast Feeding within the first hours of delivery were 134 out of 218, this should be 61.47% and not 61.8% as reported. (Also choose to use either one or two decimal places throughout the paper).

• There was no mention of husbands but close friends in the results section, authors’ conclusion however reads “disclose their test results to their husbands”. Conclusion should reflect the results.

Background

• Authors need to have a brief but detailed synopsis including statistics of what is known about factors influencing EBF among HEIs particularly in the LMICs including sub-Saharan Africa, what’s not known and the gaps to be addressed by the study. This study’s background lacks such information.

Materials and Methods

• Page 5: Authors claimed to have used a mixed method approach and that qualitative data were used to triangulate the quantitative findings; while this could be true, this paper doesn’t reflect such.

Results

• Page 8-9: The average age is 28.68 in the abstract and 28.6 is the main results section, also SD is reported as ± 4.2 in the abstract and 4 in the results. Authors should check for consistency.

• Page 9: Can the authors include p-values and Odds Ratio in Table 1

• Page 10: Authors claimed that findings from the FGDs supported the ‘knowledge factor’. Their analysis didn’t include any qualitative analysis, additionally, qualitative data doesn’t feature anywhere in the results section.

• Page 10: By stating that majority of HIV positive mothers are not willing to practice EBF because of fear of transmission of HIV through breast milk, but some of them who got advice of health professionals adhere to EBF; this being a key finding, authors should support the same with statistical data (OR, AOR, CI)

• Page 10-11: Table 2 &3: should include inferential statistics as well

• Page 12: Authors have mentioned that they controlled for effect of confounders; they should give a brief description of the statistical model(regression) used to eliminate the confounders [stratification or multivariate methods] at the methods section. Also describe which covariates were controlled, as shown in the descriptive analysis tables 1,2&3

• Page 12: Table 4 - Include p-values in the table or * to follow the OR values; then have a key below the table with asterisks like *p<0.05. **p<0.01. ***p<0.001

Discussion

• Page 13: Mothers who disclosed their HIV status to their close friends were 3 times more likely to adhere to EBF for the first six months than their counter parts. Please check for consistency, is it 3 or 6 times?

• Page 14: The quote by the 29-year-old mother should be in the results. The study mentions qualitative methods using FGD but the findings are entirely quantitative with a tacit mention of the qualitative findings in the results' section. The author should analyze and report the qualitative data to support the triangulation mentioned in the methods. Or if the qualitative results have been published elsewhere, this should be indicated

• Page 14: On strength of the study; authors stated that their findings were triangulated by qualitative data; This is not true. The author did not include a single qualitative quote in the results. Secondly, there is no mention of how the qualitative data was analyzed from coding to theme development.

Minor comments

• Participants were HIV positive mothers; were participants given the option not to participate without any fear of discrimination? like service denial at the health facilities.

• Conclusion should also include contributions to existing literature, significance of results and conclude with your thoughts.

Reviewer #2: Dear Editor

Review of Manuscript PONE-D-21-08205

Thank you for the opportunity to review this interesting manuscript. I am of the view that this manuscript is important given that the authors address an important aspect of breastfeeding for a vulnerable population. Following review of the manuscript, please find my comments as follows.

Comments to the authors

The methods used are technically sound with a fair presentation of quantitative data. The statistical analyses used is adequate for the objectives of the study. However, the quantitative data presented has errors that need to be addressed. The qualitative data presented is also sparse and the paper would benefit from some more qualitative data content. Importantly, the manuscript needs extensive grammatical reconstruction and I recommend that it would benefit from proofreading and copyediting by a high-proficiency English speaker.

The following are detailed comments and recommendations

Title

I recommend the title undergo a grammatical edit. May I suggest the full stop be replaced with a colon i.e. “…western Ethiopia: a cross-sectional study.”

Abstract

The abstract, like the rest of the paper would benefit from extensive proofreading and copyediting preferably by a highly proficient English speaker.

Background

Page 3, paragraph 2: The linkages between the cost of replacement feeding, poor water and sanitation access and EBF recommendations needs to be more explicitly explained in the background section.

Page 4, paragraph 2 & 3: These paragraphs are missing some citations. The entire paragraph, despite citing several findings only cites one publication.

Methods

The methods section is sufficiently detailed with explicit sampling and data collection details. The methods are technically sound and suitable for the objectives of the study.

Results

I recommend that the authors maintain a consistent number of decimal places in the tables presented.

The authors must review the tabular data presented as some of the frequencies and percentages do not add up or tally. Eg Table 2 “No of ANC follow up” and Table 3 “When starts breast feed”

The results section has very little qualitative data representation. I recommend that the results section would be made more interesting with the inclusion of additional quotes from qualitative responses. I recommend that quotes regarding reasons for time of giving colostrum and how HIV status disclosure affected breastfeeding would be interesting and give support to the quantitative results. This would also strengthen the claim of triangulation cited in the strengths section of the manuscript.

I hope that the authors will consider these suggestions in improving the manuscript to address this very important topic. I wish them all the best in this endeavour.

6. PLOS authors have the option to publish the peer review history of their article (what does this mean?). If published, this will include your full peer review and any attached files.

Reviewer #1: **Yes: **Jefferson Mwaisaka

Reviewer #2: No

---

## [Author Response · Author response to Decision Letter 0]

14 Dec 2021

Response to Reviewers 

Title: Factors influencing Exclusive Breast Feeding among children born to HIV positive mothers attending public health facilities in western Ethiopia: Cross-sectional study.

Authors: Ejigayehu Tolessa Bultum (ebultum@yahoo.com

 Elias Merdassa Roro (emerdassa@gmail.com)

 Tsedeke Wolde (tsedekewolde@yahoo.com)

 Ilili Feyissa Regassa (liasfeyisa2@gmail.com) 

Version: 2 Date: 05 November 2021 

Author’s Response to Reviewers Comments: See over 

Editor’s Comments 

Version: 2 

Date: 05 November 2021

Title: Factors influencing Exclusive Breast Feeding among children born to HIV positive mothers attending public health facilities in western Ethiopia: Cross-sectional study.

Editor’s comment -1 

1. Please ensure that your manuscript meets PLOS ONE’s style requirements, including those for file naming. 

Authors Response -1

1. Thank you for the comment. We have now the modified the revised manuscript following PLOS ONE’s style requirements, including those for file naming. 

Editor’s comment -2

1. Please include additional information regarding the survey or questionnaire used in the study and ensure that you have provided sufficient details that others could replicate the analyses. For instance, if you developed a questionnaire as part of this study and it is not under a copyright more restrictive than CC-BY, please include a copy, in both the original language and English, as Supporting Information. 

Authors Response -2

1. We have included additional information in the revised manuscript sufficient details regarding the questionnaire that others could replicate the procedures for development and analysis. But we want to clear out that, this questionnaire is not standardized and, not been validated by other scholars or organizations/institution. Hence, we can share the English version of the questionnaire with you or any other researchers who want to check when asked through our email address on the manuscript. But it may not be appropriate to share it publicly, which may mislead other researchers as this questionnaire might have limitations. This may lead other to repeat same errors as we might have done, especially early career researchers and students. In the revised manuscript, we have provided additional detail information regarding all the procedures, contents of the questionnaire in ‘Data Collection Procedures’ section of the main text of the revised manuscript (line 19-30 on page 6, and line 1-4 on page 7), now read as follows; 

Data Collection Procedures 

‘The interview questionnaire was prepared by reviewing relevant literature as part of this study design. The interview questionnaire contained eight (8) sections, each addressing different issues. The questionnaire was first developed in English and, translated into the local language (Afaan Oromo) then, back translated to English to check for its consistency by language experts who can understand both languages. The tool was pre-tested among breast feeding mothers to check language clarity, administration procedures and consistency in a similar setting to the study site, but outside of the study catchment area. After pre-testing, data were analyzed and carefully examined for any discrepancy between actual meaning and respondents’ answers. After thorough discussion between investigators, data collectors, and supervisors, some ambiguous words were modified and misunderstandings were clarified. Data from the pre-test are not included in this manuscript. 

The data in this study were collected by 13 ART nurses from the respective health institutions who were working at ART and PMTCT department. Three supervisors and principal investigator performs overall controlling activities of data collections process. Two days intensive training was provided for data collectors and supervisor outlining data collection process and tools.’

Editor’s comment -3

1. You indicated that you had ethical approval for your study. In your Methods section, please ensure you have also stated whether you obtained consent from parents or guardians of the minors included in the study or whether the research ethics committee or IRB specifically waived the need for their consent. 

Authors Response -3

1. Thank you for your insightful comment. As you have commented, we have now inserted a statement highlighting the consent obtaining procedures where minors might be involved. It is now appearing the main text of the revised manuscript under ‘Ethics approval and consent to participate’ Section (line 8-20, on page 8), now reads, 

‘Ethics approval and consent to participate

 This study was approved by the Wollega University, Institute of Health Sciences, Research Ethics Committee. Letters of permission were also obtained from East, West and Kellem wollega zones health offices and the respective health offices and facilities of the three selected districts and kebeles. A consent form was attached to every questionnaire with brief information sheet explaining the main aims of the study, benefits, and risks of the study to the participants, explaining their full right to withdraw at any time and their right to skip any question that they did not want to answer. Written consent was obtained from each participant. Data were collected after each participant signed a consent form. In cases where data were required from minors, consent was sought from their parents or guardians. Only those who consented participated in the study. Data were collected at each selected health facility (i.e., hospital or health centers) in an area providing maximum privacy. All individual identifiers were removed to maintain anonymity of participants and each questionnaire was assigned a unique number.’ 

Editor’s comment 4 

1. Please ensure you have thoroughly discussed any additional potential limitations of this study within the discussion section, including the potential impact of confounding factors. 

Authors Response -4

1. In the revised manuscript we have inserted short paragraph on potential impacts confounders in the entry paragraph of discussion section (line 1-4, on page 14). Now it appears as follows; 

‘Our study had some limitations. Our findings need to interpreted with causation as other confounding factors may have impacted when establishing associations as we considered only those variables with P-values <0.05 at bi-variable analysis. We also had limited variables included in the final model.’

Editor’s comment - 5 

1. In your Data Availability statement, you have not specified where the minimal data set underlying the results described in your manuscript can be found. PLOS defines a study's minimal data set as the underlying data used to reach the conclusions drawn in the manuscript and any additional data required to replicate the reported study findings in their entirety. All PLOS journals require that the minimal data set be made fully available. For more information about our data policy, please see http://journals.plos.org/plosone/s/data-availability. 

"Upon re-submitting your revised manuscript, please upload your study’s minimal underlying data set as either Supporting Information files or to a stable, public repository and include the relevant URLs, DOIs, or accession numbers within your revised cover letter. For a list of acceptable repositories, please see http://journals.plos.org/plosone/s/data- availability#loc-recommended-repositories. Any potentially identifying patient information must be fully anonymized. 

Authors Response -5

Thank you for the comment. In the revised manuscript, we have now provided detail information regarding data availability under ‘Availability of data and materials’ section (line 1-6 on page 15). It reads as follow;

‘All data used for the write up of this manuscript are fully available in SPSS readable formats. Results are fully presented in tables, narrations and figures in this manuscript, nothing is left unpresented. The sampling frame with the lists of every respondent is also available in Excel readable format. Data has not been uploaded to a public repository for confidential and ethical reasons.’ 

Editor’s comment 6

1. Your ethics statement should only appear in the Methods section of your manuscript. If your ethics statement is written in any section besides the Methods, please delete it from any other section. 

Authors Response -6

1. We have removed ethics statement elsewhere than methods section of the revised main text manuscript. Not it only appears under ‘Ethics approval and consent to participate’ (line 8-20 on page 8). 

Editor’s comment- 7 

1. We suggest you thoroughly copyedit your manuscript for language usage, spelling, and grammar. If you do not know anyone who can help you do this, you may wish to consider employing a professional scientific editing service. 

A copy of your manuscript showing your changes by either highlighting them or using track changes (uploaded as a *supporting information* file).

Authors Response -7 

1. The revised manuscript has been copy edited for language usage, spelling and grammar by health professional from England by a native English speaker. The editor had a rich experience in editing manuscripts. The editor detail information is included in the acknowledgments section of the revised manuscript. 

2. We have uploaded the copy-edited manuscript with track change as *supporting inflation* file

Review Comments to the Author 

Title: Factors influencing Exclusive Breast Feeding among children born to HIV positive mothers attending public health facilities in western Ethiopia: Cross-sectional study.

Reviewer #1: General comments

• This is a good topic and will be of great interest to reader

• The manuscript needs to be edited for grammar and syntax (by a native English speaker) 

General comments: Authors Response #1

• Thank for your appreciations and professional interest of the article. 

• The revised manuscript is now edited for grammar and syntax by native English speaker and health professional. 

Major comments-Review #1

• Topic: Status of Exclusive Breast Feeding among children born to HIV positive mothers 

attending public health facilities in western Ethiopia. Cross-sectional study.

• This study is more of Exploring factors associated with or influencing EBF among 

• children born to HIV positive mothers and not status of EBF as stated by the authors 

Major comments: Authors Response to Reviewer #1 

• Topic: Your comment and suggestion are well accepted. We have now modified the title in line with your suggestion. But still, we are open to take any further suggestion for modification. For now, in the revised manuscript, the modified topic reads as follows;

 ‘Factors influencing Exclusive Breast Feeding among children born to HIV positive mothers attending public health facilities in western Ethiopia: Cross-sectional study.’

Abstract: Review #1 Comments

• Page 2: The abstract of this paper should contain a statement on the background of EBF among HEIs in Ethiopia. This is missing.

• The purpose the of the study is not consistently stated throughout the paper. The authors claim to be assessing the levels and factors influencing EBF among HEIs while the topic reads Status of EBF. The study should have one clear focus and purpose that’s carried out throughout.

• The proportion of respondents who initiated Exclusive Breast Feeding within the first hours of delivery were 134 out of 218, this should be 61.47% and not 61.8% as reported. (Also choose to use either one or two decimal places throughout the paper).

• There was no mention of husbands but close friends in the results section, authors’ conclusion however reads “disclose their test results to their husbands”. Conclusion should reflect the results.

Abstract :Authors Response to Reviewer #1 

• Page 2: The abstract of this paper now contains a statement on the background of EBF among HEIs in Ethiopian context. In the revised manuscript, now we have inserted background statement integrated with objective of the study (line 2-6, on page 2), which reads as follows,

‘Only about 39% of infants in the developing countries are exclusively breast-fed for the first six months. Human immunodeficiency virus (HIV) positive women were confused about feeding methods. Exclusive Breastfeeding (EBF) practice of Human immunodeficiency virus (HIV) positive mothers is sub-optimal in Ethiopia. Hence, we want to identify the main factors influencing exclusive breast-feeding among HIV positive breast-feeding mothers.’ 

• Thank you for the comment. We have come to consensus that as you have commented above regarding the title of the article, this comment is also related with the comment above. We have modified title and we have tried to keep the consistency of the article with specific focus with factors influencing EBF among HIV positive mothers with infants less than six months old. Otherwise, we are happy to hear from you with further details, if we could not get the main idea of your comment.

• Great insight, very important comment. It was really an error. Now we have corrected it to 134(61.5%) rounded to the closed decimal point and highlighted in the revised main text manuscript. We have tried to stick to a ‘uniform decimal points’ (we have chosen single decimal point) throughout the paper including data in tables. Now it appears in the main text of revised manuscript (line 3, on page 11). 

• Thank you for this critical comment. It was a mistake as well. The mistake emerged from the questionaries (i.e., in the questionnaire we administered to the participants had five choices). Later in the multivariable table 4, we have recoded those five choices into dichotomous variables i.e., ‘YES/NO’. We forgot to change it in the submitted version of the manuscript. Now in the revised manuscript, we have modified/corrected across all sections of the manuscript (i.e.; results, discussion and conclusion) into same/uniform expression. Now it reads;

‘disclosure of HIV status to someone close to them including their husband’

Background: Review #1 Comments 

• Authors need to have a brief but detailed synopsis including statistics of what is known about factors influencing EBF among HEIs particularly in the LMICs including sub-saharan Africa, what’s not known and the gaps to be addressed by the study. This study’s background lacks such information. 

Background: Authors Response to Reviewer #1 

• We had some before. We have added more literatures. We are not sure if it is sufficient. 

Materials and Methods: Review #1 Comments 

• Page 5: Authors claimed to have used a mixed method approach and that qualitative data were used to triangulate the quantitative findings; while this could be true, this paper doesn’t reflect such. 

Materials and Methods: Authors Response to Reviewer #1.

• Dear reviewers, we thank you for your very valuable comment regarding qualitative part of our manuscript. Both reviewers have commented on this issue. We are convinced that, qualitative part of this manuscript is very shallow. We came to consensus that qualitative data to be removed from this manuscript but we have data on hand. We will try to re-analyze and submit it independently for another time. For now, we have removed all qualitative data from this manuscript. 

Results: Review #1 Comments

• Page 8-9: The average age is 28.68 in the abstract and 28.6 is the main results section, also SD is reported as ± 4.2 in the abstract and 4 in the results. Authors should check for consistency.

• Page 9: Can the authors include p-values and Odds Ratio in Table 1

• Page 10: Authors claimed that findings from the FGDs supported the ‘knowledge factor’.

• Their analysis didn’t include any qualitative analysis, additionally, qualitative data doesn’t feature anywhere in the results section.

• Page 10: By stating that majority of HIV positive mothers are not willing to practice EBF because of fear of transmission of HIV through breast milk, but some of them who got advice of health professionals adhere to EBF; this being a key finding, authors should support the same with statistical data (OR, AOR, CI)

• Page 10-11: Table 2 &3: should include inferential statistics as well.

• Page 12: Authors have mentioned that they controlled for effect of confounders; they should give a brief description of the statistical model(regression) used to eliminate the confounders [stratification or multivariate methods] at the methods section. Also describe which covariates were controlled, as shown in the descriptive analysis tables 1,2&3

• Page 12: Table 4 - Include p-values in the table or * to follow the OR values; then have a key below the table with asterisks like *p<0.05. **p<0.01. ***p<0.001. 

Results: Authors Response to Reviewer #1. 

• Good comment. We appreciate for that. Page 8-9: In the revised manuscript the average age is now corrected to 28.6 in the abstract and 28.6 is the main results section, also SD is corrected as ± 4 in the abstract and 4 in the results respectively. We Authors have now checked for consistency. Now it is consistent. 

• Page 9: Now we have inserted p-values across tables (1,2 ,3). We appreciate the concerns across the tables regarding P-values and Odds Rations. Tables 1,2, & 3 are just descriptive tables and intends to present frequency of variables only (i.e., descriptive statics tables). Though we were not interested to see a sort of statistical associations (i.e., inferential statistics), here have presented P-values for the sake of your comment, but not Odds Rations here. We also did not find in PLOSONE guidelines and publications as well. We have presented statistical associations (inferential statistics) in table 4. 

• Page 10: We will re-analyze it and make separate article rather than using for triangulation for another submission. Hope you will help us more by then. 

• Page 10: Thank you for your comment. This statement was taken from Qualitative finding (FGD) summary finding for triangulation purpose. As far as we know, we cannot stablish statistical associations (OR<AOR, CI) from qualitative data. Now the qualitative data were removed from the revised manuscript for the reason we have stated above. 

• Page 10-11: Although we have no interest in inferential statistics in tables 1,2,3. We have presented present p-values in tables 1, 2 & 3. We have summarized ‘COR, AOR and CI’ in table 4. 

• Page 12: We have briefed already the method we used for controlling for cofounders’ methods section of the main text of the manuscript. Basically, we run ‘multi-variable logistic regression model which was used to control for the effects of cofounders on the outcome variable.’ This phrase comes from the main text manuscript. 

• Page 12: In table 4, we have presented AOR with 95% CI. We have followed your suggestion to use ‘*’ to follow the OR values; then have a key below the table with asterisks like *p<0.05. **p<0.01. ***p<0.001. 

Discussions: Review #1 Comments 

• Page 13: Mothers who disclosed their HIV status to their close friends were 3 times more likely to adhere to EBF for the first six months than their counter parts. Please check for consistency, is it 3 or 6 times?

• Page 14: The quote by the 29-year-old mother should be in the results. The study mentions qualitative methods using FGD but the findings are entirely quantitative with a tacit mention of the qualitative findings in the results' section. The author should analyze and report the qualitative data to support the triangulation mentioned in the methods. Or if the qualitative results have been published elsewhere, this should be indicated 

• Page 14: On strength of the study; authors stated that their findings were triangulated by qualitative data; This is not true. The author did not include a single qualitative quote in the results. Secondly, there is no mention of how the qualitative data was analyzed from coding to theme development. 

Discussions: Authors Response to Reviewer #1

• Page13: Thank your insightful comment. On page 13, it was an error. Now we have corrected to 6 times, not 3 times. Now it consistent across the revised manuscript. 

• Page 14: Thank you for the comment again. Now we have agreed to remove the qualitative issue from this revised manuscript. We may come back with an independent manuscript with qualitative data only. 

• Page 14: We appreciate your concerns. We have removed qualitative issues across the entire manuscript. In the revised manuscript, we have modified the strength section. 

Minor comments: Review #1 Comments

• Participants were HIV positive mothers; were participants given the option not to participate without any fear of discrimination? like service denial at the health facilities.

• Conclusion should also include contributions to existing literature, significance of results and conclude with your thoughts. 

Minor comments: Authors Response to Reviewer #1

• Yes, definitely, in the ethics sections, we have described in detail regarding participants full right to with draw at any time from participation. We presented here again the ethics statements as follows. 

‘‘Ethics approval and consent to participate

This study was approved by the Wollega University, Institute of Health Sciences, Research Ethics Committee. Letter of permission was also obtained from East, West and kellem wollega zones Health Offices and respective health offices and facilities of the three selected districts and kebeles. Consent form was attached to every questionnaire with a brief information regarding the main aim of the study, benefits, and risks of the study to the participants, explaining their full right to withdraw at any time, skip any question that they don’t want to answer. Written consents were obtained from each respondent. Data were collected after each respondent signed a consent form based on their full consent. In a case data were required from minors, consents were sought from their parents or guardians. Only those who consented participated in the study. Data were collected at each selected health facility (i.e., Hospital or Health centers) level in an area which gives them the maximum privacy, after getting services. We removed an individual identifier to maintain the anonymity of the respondents by assigning each questionnaire a unique number.’

• Thank you for the comment. In the revised manuscript, we have now modified the conclusion section. 

Review comments forwarded from reviewer #2 and authors response to the reviews. 

Title: Factors influencing Exclusive Breast Feeding among children born to HIV positive mothers attending public health facilities in western Ethiopia: Cross-sectional study.

Reviewer #2: Comments to the authors

• The manuscript needs extensive grammatical reconstruction and I recommend that it would benefit from proofreading and copyediting by a high-proficiency English speaker.

• Title

I recommend the title undergo a grammatical edit. May I suggest the full stop be replaced with a colon i.e. “...western Ethiopia: a cross-sectional study.” 

Authors Response to Reviewer #2

• Thank for the comment. The revised manuscript is now copy edited for grammatical errors by health professional native English speaker from England. 

• Title grammatical edit. We accept the comment. We have now incorporated your suggestion regarding the replacement of full stop with colon. The title is also modified based on the suggestion from the other reviewer as follows. 

‘Title: Factors influencing Exclusive Breast Feeding among children born to HIV positive mothers attending public health facilities in western Ethiopia: Cross-sectional study.’

Abstract: Reviewer #2 Comments to the authors

• Abstract

The abstract, like the rest of the paper would benefit from extensive proofreading and copyediting preferably by a highly proficient English speaker. 

Abstract: Authors Response to Reviewer #2

• Abstract: The revised manuscript including abstract section is also now benefited from the overhaul copy edited for grammatical errors by health professional native English speaker from England. The profile of the editor is found in the acknowledgments section of the revised manuscript. 

Background: Reviewer #2 Comments to the authors

• Page 3, paragraph 2: The linkages between the cost of replacement feeding, poor water and sanitation access and EBF recommendations needs to be more explicitly explained in the background section.

• Page 4, paragraph 2 & 3: These paragraphs are missing some citations. The entire paragraph, despite citing several findings only cites one publication.

Background: Authors Response to Reviewer #2

• In the revised manuscript, we have now explained on Page 3, paragraph 2: the linkages between the cost of replacement feeding, poor water and sanitation access and EBF recommendations in the background section. 

• In the revised manuscript we have included the missing citations and rearranged the context the paragraphs during proof reading and copy edition. We have checked for the missing citations and other contexts on Page 3 & 4, paragraph 2 & 3. 

Methods: Reviewer #2 Comments to the authors

• The methods section is sufficiently detailed with explicit sampling and data collection details. The methods are technically sound and suitable for the objectives of the study.

Methods: Authors Response to Reviewer #2

• Thank you very much for your appreciations and encouragements. 

Results: Reviewer #2 Comments to the authors

• I recommend that the authors maintain a consistent number of decimal places in the tables presented.

• The authors must review the tabular data presented as some of the frequencies and percentages do not add up or tally. Eg Table 2 “No of ANC follow up” and Table 3 “When starts breast feed”

• The results section has very little qualitative data representation. I recommend that the results section would be made more interesting with the inclusion of additional quotes from qualitative responses. I recommend that quotes regarding reasons for time of giving colostrum and how HIV status disclosure affected breastfeeding would be interesting and give support to the quantitative results. This would also strengthen the claim of triangulation cited in the strengths section of the manuscript.

Results: Authors Response to Reviewer #2

• Thank you for the comments, this is relevant for enrichment of data presentations. We have maintained uniform/consistent decimal point (one decimal points) presentations throughout the revised main text manuscript.

• Tabular data presentations. We have carefully cross-checked the discrepancy and addressed them accordingly in the revised manuscript. We have highlighted the changes in the corresponding tables. I.e. Table 2 “No of ANC follow up” and Table 3 “When starts breast feed”

• Though the qualitative data is very important for the enrichment of this manuscript, for this manuscript, we have removed qualitative data because of we felt that we could not address it very well here. We may come back with separate manuscript exclusively with qualitative data.

---

## [Decision Letter · Decision Letter 1]

20 Jan 2022

PONE-D-21-08205R1Factors influencing Exclusive Breast Feeding among children born to HIV positive mothers attending public health facilities in western Ethiopia: Cross-sectional study.PLOS ONE

Dear Mr Elias Roro,

Thank you for submitting your manuscript to PLOS ONE. After careful consideration, we feel that it has merit but does not fully meet PLOS ONE’s publication criteria as it currently stands. Therefore, we invite you to submit a revised version of the manuscript that addresses the points raised below during the review process.

We look forward to receiving your revised manuscript.

Kind regards,

Professor Kwasi Torpey, MD PhD MPH

Academic Editor

PLOS ONE

Journal Requirements:

Reviewers' comments:

Reviewer's Responses to Questions

**Comments to the Author**

1. If the authors have adequately addressed your comments raised in a previous round of review and you feel that this manuscript is now acceptable for publication, you may indicate that here to bypass the “Comments to the Author” section, enter your conflict of interest statement in the “Confidential to Editor” section, and submit your "Accept" recommendation.

Reviewer #2: (No Response)

2. Is the manuscript technically sound, and do the data support the conclusions?

Reviewer #2: Partly

3. Has the statistical analysis been performed appropriately and rigorously? 

Reviewer #2: Yes

4. Have the authors made all data underlying the findings in their manuscript fully available?

Reviewer #2: Yes

5. Is the manuscript presented in an intelligible fashion and written in standard English?

Reviewer #2: No

6. Review Comments to the Author

Reviewer #2: Dear Editor

Review of Manuscript PONE-D-21-08205

Comments to the authors

I appreciate the work put in by the authors to address my earlier comments and incorporate recommendations.

Again I would like to stress the importance of having this manuscript professionally copyedited and proofread due to existing grammatical errors in the script.

The following are detailed comments and recommendations

Results

The authors must again review the tabular data presented as some of the frequencies and percentages are wrong.

The authors would either have to include the qualitative quotes in the results section or drop their claim of using triangulation.

I hope that the authors will consider these suggestions in improving the manuscript to address this very important topic. I wish them all the best in this endeavour.

7. PLOS authors have the option to publish the peer review history of their article (what does this mean?). If published, this will include your full peer review and any attached files.

Reviewer #2: No

---

## [Author Response · Author response to Decision Letter 1]

3 Apr 2022

Response to Reviewers 

Title: Factors influencing Exclusive Breast Feeding among children born to HIV positive mothers attending public health facilities in western Ethiopia: Cross-sectional study.

Authors: Ejigayehu Tolessa Bultum (ebultum@yahoo.com

 Elias Merdassa Roro (emerdassa@gmail.com)

 Tsedeke Wolde (tsedekewolde@yahoo.com)

 Ilili Feyissa Regassa (liasfeyisa2@gmail.com) 

Version: 3 Date: 03 April 2022 

Author’s Response to Reviewers Comments: See over 

Editor’s Comments 

Version: 3 

Date: 03 April 2022

Title: Factors influencing xxclusive breast feeding among children born to HIV positive mothers attending public health facilities in western Ethiopia: Cross-sectional study.

Editor’s comment -1 

1. Again, I would like to stress the importance of having this manuscript professionally copyedited and proofread due to existing grammatical errors in the script. 

Authors Response -1

1. Thank you for the comment. The revised manuscript has been copy edited for language usage, spelling and grammar by health professional from Australia by a native English speaker. The editor had a rich experience in editing manuscripts. The editor is currently post-doctoral fellow at the university of Queensland Mater research institute. 

Results 

Editor’s comment -1

1. The authors must again review the tabular data presented as some of the frequencies and percentages are wrong. 

2. The authors would either have to include the qualitative quotes in the results section or drop their claim of using triangulation. 

Authors Response -1

1. This is very relevant comment. In the revised manuscript we have revised our tables across the document. We have now revised some of the frequencies and percentages in table 2,3 & 4 in the revised manuscript. We indicated those changes using track and changes. We have now the manuscript with those changes labelled and uploaded as ‘manuscript with track change’ document. 

2. We have already removed the qualitative section and dropped the triangulation claim from the manuscript. 

3. We have also modified the Abstract section. This modification is also indicated in modified manuscript labeled as “manuscript with track changes”

It reads as follows: 

‘In our study, only just over half of the mothers practiced EBF for the first six months. Care providers should continue to encourage mothers to practice EBF in the first six months and to disclosure of HIV status to someone close to them including their husband. Efforts should be in place to curb the risk of HIV/AIDS transmission during delivery. Continues advise for mothers to practice EBF in the first 6 months is still needed.’

---

## [Editor Report · Decision Letter 2]

27 Jun 2022

Factors influencing exclusive breast feeding among children born to HIV positive mothers attending public health facilities in western Ethiopia: Cross-sectional study.

PONE-D-21-08205R2

Dear Mr Elias Roro,

We’re pleased to inform you that your manuscript has been judged scientifically suitable for publication and will be formally accepted for publication once it meets all outstanding technical requirements.

Kind regards,

Professor Kwasi Torpey, MD PhD MPH

Academic Editor

PLOS ONE

Additional Editor Comments (optional):

Copyediting done but a few typos remain which can be addressed during production
---

## [Editor Report · Acceptance letter]

14 Jul 2022

PONE-D-21-08205R2 

Factors influencing exclusive breast feeding among children born to HIV positive mothers attending public health facilities in western Ethiopia: Cross-sectional study. 

Dear Dr. Roro:

I'm pleased to inform you that your manuscript has been deemed suitable for publication in PLOS ONE. Congratulations! Your manuscript is now with our production department. 

Kind regards, 

on behalf of

Dr. PLOS Manuscript Reassignment 

Staff Editor

PLOS ONE